# Impacts of Land Ownership on the Economic Performance and Viability of Rice Farming in Thailand

**Amorn Pochanasomboon, Witsanu Attavanich \*and Akaranant Kidsom**

Department of Economics, Kasetsart University, 50 Phahonyothin Rd, Chatuchak, Bangkok 10900, Thailand; amorn.p@ku.th (A.P.); fecoanki@ku.ac.th (A.K.)

\*   Correspondence: fecowna@ku.ac.th

**Abstract:** This article evaluates the impacts of land ownership on the economic performance and viability of rice farming in Thailand, and explores whether they are heterogeneous across different types of farming while using the propensity score matching (PSM) technique. This study categorizes land ownership into two types: full land ownership and weak land ownership. We reveal that full land ownership enhances the rice yield of small and midsize farms, with values of 115.789–127.414 kg/hectare and 51.926–70.707 kg/hectare, respectively. On the other hand, weak land ownership only enhances the rice yield of small farms, with an increased yield of 65.590–72.574 kg/hectare. Full land ownership also helps to reduce the informal debt of small and midsize farms by $16.972–$24.877 per farm and $31.393–$37.819 per farm, respectively. On the other hand, weak land ownership helps to reduce the informal debt of midsize farms, ranging from $36.909 to $44.681 per farm. Therefore, policy makers should encourage small and midsize farm households to adopt full land ownership instead of weak land ownership, as this will provide the greatest benefits to farm households and efficient land use.

**Keywords:** land ownership; rice farming; rice yield; informal debt; propensity score matching; Thai agriculture

## 1. Introduction

Over the past few decades, the increasing global population, changing climate, and antitrade policies have challenged the agricultural sector and, hence, the global food security [1,2]. Thailand has been one of the world's major food-producing and -exporting countries, and the agricultural sector has played a crucial role in both economic and social aspects. It employed as much as 32.2 percent of the country's labor force, covering 6.4 million households, and it generated income from exporting agricultural products of approximately 253 billion US dollars in 2018 [3,4]. It also acts as a buffer to alleviate the unemployment problem that was created by the economic crisis and contributed 9.3 percent of the gross domestic product in 2017 [5].

Among agricultural commodities, rice is the most important one in Thailand. Not only is it a major staple food of the Thai and global population, but it is also a major source of the country's export revenue. According to [6], Thailand was the world's second largest rice exporter in 2018, with the export quantity of 11.1 million tons generating the export revenue of 5,571 million US dollars, contributing 11.4 percent of the global export value. Moreover, the area of rice production accounted for 46 percent of the country's total cropland in 2017, with 4.2 million farm households in 2019 [7], reflecting the importance of rice production in the Thai agricultural sector.

While the rice production in Thailand has provided several benefits, there are several uncontrolled factors affecting rice production and, thus, the well-being of rice farmers, as well as global food security. These uncontrolled factors include, for example, increased costs of production, climate variability, and natural disasters. Besides the above factors, several other uncontrolled factors that come from internal farm management also affect rice production, such as a lack of knowledge in applying modern technology, incomplete knowledge of financial management, and a mismatch between crops and cropland suitability [8]. These uncontrolled factors have obstructed improvement in the efficiency of rice production, consequently resulting in a food security concern at the global level, because the food supply cannot meet the escalating food demand. At the national level, these factors have reduced the competitiveness of rice in the global market and caused the problem of agricultural household debts, which might lead to several social problems.

According to [9], the rice yield in Thailand has remained stagnant for over a decade and it is lower than the rice yield in several major rice-producing countries (i.e., China, Vietnam, Bangladesh, Philippines, India, Myanmar, and Cambodia). When considering the agricultural household debt, the informal debt has continually increased over the past two decades, thus leading to the loss of farmland due to the dramatically high interest rate.

Several previous studies revealed that an increase in land ownership is a key solution to promote the food security, efficiency of rice production, and economic viability of rice farming [10–12]. With a well-defined property right structure, rational individuals will use their resource efficiently because a decline in the value of the resource represents a personal loss [13]. In Thailand, reference [14] revealed that only 59.3 percent of agricultural households fully owned land, which is consistent with the well-defined property right structure in 2017. Studies have also revealed that the size of a farm also statistically influences the economic performance and viability of farming [11,15].

Although there are several previous studies investigating the effects of land ownership on the economic performance and viability of farming in several countries [10–12,15,16], research studies in Thailand are limited and there have been no studies empirically exploring these issues in Thailand since [10,16] that evaluated the impact of land ownership in three out of 77 provinces of Thailand. To our knowledge, there is no study that has investigated the effect of land ownership at the country level with farm-level data and exploring the heterogeneous impact of land ownership across farm types.

Therefore, the objectives of this study are to evaluate the impact of land ownership on the economic performance and viability of rice farming and explore whether the effects of land ownership are heterogeneous across different types of farm, including small, midsize, and large farms. Propensity score matching (PSM) techniques that were first introduced by [17] are used to address the possible self-selection bias of the constructed farm-level dataset that was obtained from various sources. We split land ownership into two types to deeply understand the impact of land ownership. The first type, named "full land ownership", captures all the characteristics of the well-defined property right structure in economic theory, consisting of exclusivity, transferability, and enforceability, while the second type, named "weak land ownership", only captures the characteristics of exclusivity and enforceability. In brief, exclusivity implies that all of the benefits and costs should only accrue to the owner. Transferability implies that property rights should be transferred to others, and enforceability implies that property rights should be secure from encroachment [18]. Details are provided in the section presenting the materials and methods.

This article is organized, as follows: Section 2 presents details of materials and methods used for the analysis; Section 3 provides results; Section 4 discusses the findings; and, Section 5 presents the conclusions and policy implications that were drawn from the findings.

## 2. Materials and Methods

### 2.1. Methods

In experimental approaches, treatment assignment can be randomized and, therefore, a comparison of potential outcomes for the treated and control groups can provide statistically valid

estimates of treatment effects. However, a farm with land ownership is not random due to the voluntary nature of the farmland owner's choice. An estimation of the effect of land ownership might be confounded by the possible correlation between economic outcomes and factors affecting the decision to own land.

This study applies a framework with two potential outcomes to overcome the problem of self-selection bias: $Y^1$—an outcome of farms with land ownership (treated farms)—and $Y^0$—an outcome of farms without land ownership (control farms). The observed outcome for any individual farm *i* can be written as $Y_i = T_i \cdot Y_i^1 + (1 - T_i) \cdot Y_i^0$, where $T \in \{0,1\}$ indicates the treatment status, with $T = 1$ if a farm has land ownership. The gain/loss of an individual farm *i* with land ownership is $\Delta_i = Y_i^1 - Y_i^0$. Estimating the individual farm treatment effect $\Delta_i$ is not possible and we have to concentrate on (population) average treatment effects (ATE) because we cannot observe both outcomes for an individual farm, as shown in Equation (1) [19].

$$ATE = E(Y_i^1 - Y_i^0) \tag{1}$$

The most widely used evaluation parameter is the "average treatment effect on the treated" (ATT), which, in our context, represents the difference between the expected economic performance and viability outcomes of farms with and without land ownership. This can be algebraically explained in Equation (2).

$$ATT = E(Y_i^1 \big| T_i = 1) - E(Y_i^0 \big| T_i = 1) \tag{2}$$

In practice, it is impossible to observe $E(Y_i^0 \big| T_i = 1)$ in Equation (2). A farm either does or does not have land ownership; treatment assignments are mutually exclusive. Estimating the ATT associated with land ownership by comparing the mean of difference for $E(Y_i^1 \big| T_i = 1)$ and $E(Y_i^0 \big| T_i = 0)$ will be erroneous due to the selection bias.

Within social science research, there are several approaches that are used to address the challenge of policy evaluation with the selection bias problem in agriculture. While using the instrumental variable, [20] analyzed the relationships between land ownership, access to finance, and female entrepreneurial performance in Eswatini, Lesotho, and Zimbabwe, and revealed that land ownership is important for female entrepreneurial performance in terms of sales levels. Using the difference-in-difference propensity score matching estimator, [21] found that agri-environmental schemes (AES) that are designed for arable land overcompensate farmers fail to comply with the World Trade Organization (WTO) rules.

An increasing number of studies [8,12,22–25] have used the propensity score matching (PSM) estimator in agricultural context to pair observations within treatment and control groups based upon the propensity score $P(X)$, which is the probability of having land ownership, by assuming that $Y^0 \perp T \big| P(X)$, where $\perp$ denotes independence. This assumption is often called the conditional independence assumption (CIA), which requires that all of the variables driving self-selection are observable to researchers [17].

It is also assumed that the probability of being treated (given covariates *X*) falls between zero and one, $0 < P(T = 1 \big| X) < 1$ to ensure overlap or common support in the distributions of all covariates *X* between farms with and without land ownership. This overlap condition ensures that overlap in the characteristics of farms with and without land ownership is sufficient for enabling proper matching. Under the CIA and overlap assumption, the PSM estimator for the ATT can be written, as shown in Equation (3):

$$\tau_{ATT}^{PSM} = E[Y_i^1 - Y_i^0 \big| T_i = 1] = E\left\{ E[Y_i^1 \big| T_i = 1, P(X)] - [Y_i^0 \big| T_i = 0, P(X)] \big| T_i = 1 \right\} \tag{3}$$

The CIA also requires the inclusion of all observed covariates *X* that simultaneously affect the probability of having land ownership and the potential outcomes in the propensity score estimation. Moreover, land ownership should not affect these variables. This study uses a combination of guidelines from economic theory, previous research studies, and statistical methods to select a set of qualified covariates *X*, as suggested by the literature [19].

We followed [26,27], by splitting the full sample into three subgroups of farm, according to the differences in the rice planted area, including small, midsize, and large farms, to evaluate whether the impacts of land ownership on the economic performance and viability of rice farming are heterogeneous across farm types and to lessen the possibility of mismatching. It is worth noting that there is no official farm typology classifying farm types in Thailand, unlike in developed countries, such as the United States and the European Union. In addition, the PSM theoretically requires large samples with substantial overlap between treatment and control groups [19,26]. We defined a small farm as having a rice planted area ≤ 1.20 hectares, midsize farm as having a rice planted area between 1.20 and 2.75 hectares, and large farm as having a rice planted area > 2.75 hectares, according to the data distribution to avoid the overlap problem. Several studies also divide the subgroups while using data distribution [8,28]. Moreover, we made several estimations by varying the cut-off points of the rice-planted area and found slightly different quantitative and qualitative results from the main findings.

This study conducted a post-matching balancing test to ensure that the covariate balancing property was satisfied. This test involves a comparison of the characteristics of farms with and without land ownership before matching and an evaluation of whether any significant differences in the characteristics of the two farm groups remain after matching. Once the post-matching balancing test was satisfied, the matching of farms with and without land ownership based on estimated propensity scores was utilized to derive the impact of land ownership on the economic performance and viability of rice farming. In addition to the imposition of common support, this study addresses the problem of limited overlap in the covariate distributions between farms with and without land ownership while using the trimming approach that was proposed by [29].

We utilized several matching algorithms as robustness checks. We firstly used nearest neighbor matching with five matching partners (NN5), ten matching partners (NN10), and kernel matching algorithms[1] because there were a large number of comparable untreated (farms without land ownership) observations in subgroups. The Gaussian kernel function was used for kernel matching. The optimal bandwidth for the kernel function was selected while using the rule of thumb that was suggested by [30]. We also used the radius matching with a caliper firstly recommended by [31][2] to increase the matching quality. However, as discussed in [32], it is difficult to know a priori what choice for the tolerance level is reasonable. We used the calipers of 0.01 and 0.02 in this study.

The quality of matching outcomes was also evaluated for each algorithm on the basis of the percent reduction of Pseudo $R^2$ and the mean standardized bias. Lastly, this study constructed two corresponding potential outcomes consisting of the rice yield and the informal debt of farm households, respectively, to capture the economic performance and viability of rice farming, which are affected by land ownership. We use STATA software version 15 for all estimation procedures.

## 2.2. Data

The household-level dataset was constructed and the main source of data was obtained from the 2013 agricultural census that was conducted by Thailand's National Statistical Office. The dataset contains 62,686 observations that were made over a crop year. After excluding the non-growing rice

---

[1]　　Kernel matching estimators are nonparametric matching estimators that use weighted averages of (nearly) all farms in the without land ownership farm group to construct the counterfactual outcome. Therefore, one major advantage of these approaches is the lower variance.

[2]　　The basic idea of radius matching is to use not only the nearest neighbour within each caliper (propensity range), but all farms without land ownership within the caliper. A benefit of this approach is that it uses only as many farms without land ownership as are available within the caliper, and thus allows for the usage of extra (fewer) units when good matches are (not) available [19].

farms, 38,980 observations remained in the dataset. Several variables used in this study were extracted and constructed from several sources, including the 2013 agricultural census, the Office of Agricultural Economics, the Royal Irrigation Department, and the Meteorological Department. The variables include the potential outcomes (i.e., rice yield and informal debt); the status of land ownership (i.e., full and weak land ownership); operator characteristics (i.e., gender, age, education level, marital status, and member of institutions status); farm characteristics (i.e., percent of agricultural labor, working in agriculture, hiring labor on a farm, source of income, ratio of rice area to total land area, rice harvested area, and integrated agriculture); and, location characteristics of farms (i.e., amount of rainfall, temperature, whether the farm is located in the municipal area, and irrigation system). Table 1 summarizes the variables that were used in the models and their definitions.

**Table 1.** Description of variables.

| Variable | Definition of Variables |
|---|---|
| *Outcome* | |
| Rice yield | Harvested rice yield (kilograms/hectare) |
| Informal debt | Amount of debt borrowed from informal financial institutions (US dollar) |
| *Treatment* | |
| Full land ownership | Whether the farm household has the land certificates consisting of the title deed and NS3 (equal to 1 if yes) |
| Weak land ownership | Whether the farm household has the land certificates consisting of the title deed, NS3, SPK401, NK, NS2, and SK1 (equal to 1 if yes) |
| *Principal characteristics* | |
| Male | Gender of the household head (equal to 1 if male) |
| Age | Age of the household head (year) |
| Primary education | Whether the household head graduated, at the least, from primary school (equal to 1 if yes) |
| Single | Whether the household head has a single marital status (equal to 1 if yes) |
| Farmer group member | Whether the household members are members of a farmer group (equal to 1 if yes) |
| Cooperative member | Whether the household members are members of farm cooperatives (equal to 1 if yes) |
| Village fund member | Whether the household members are members of a village/city fund (equal to 1 if yes) |
| Agri. assoc. member | Whether the household members are members of an agricultural association (equal to 1 if yes) |
| *Farm characteristics* | |
| Pct agri. labor | Percent of the agricultural labor to total labor in the household (%) |
| Work in agri. only | Whether the household members work only in agriculture (equal to 1 if yes) |
| Hire permanent labor | Whether the household hires permanent agricultural labor (equal to 1 if yes) |
| Hire temporary labor | Whether the household hires temporary agricultural labor (equal to 1 if yes) |
| Off-farm income | Whether the largest source of income is off-farm income (equal to 1 if yes) |
| Ratio rice area | Ratio of rice planted area to area of holding |
| Area harvested rice | Total rice harvested area (hectare) |
| Integrated agriculture | Whether the farm grows other crops or raises animals (equal to 1 if yes) |
| Small farm | Whether the farm has a rice planted area less than or equal to 1.2 hectares (equal to 1 if yes) |
| Midsize farm | Whether the farm has a rice planted area greater than 1.2 hectares and less than or equal to 2.75 hectares (equal to 1 if yes) |
| Large farm | Whether the farm has a rice planted area greater than 2.75 hectares (equal to 1 if yes) |
| *Location characteristics* | |
| Rainfall | Region-level total rainfall (millimeters) by crop year (April–March) |
| Temperature | Region-level average temperature (°C) by crop year (April–March) |
| Municipal area | Whether the farm is located in the municipal area (equal to 1 if yes) |
| Irrigate | The regional irrigated area (hectare) |

This study classifies land ownership into two types to deeply understand the role of land ownership. The first type captures, "*full* land ownership", which assumes a value of 1 if a farm household reports that he/she has land certificates that consist of the title deed and NS3. Alternatively, it takes a value of 0 if a farm household reports other types of land. The owners of the land with the title deed and NS3 have ownership of the land and they can sell the land to other people. The second type captures "*weak* land ownership", which extends the land certificates that the farm

household owns from the title deed and NS3 to SPK401, NK, NS2, and SK1. Generally, the certificates of the land tenure with SPK401, NK, NS2, and SK1 present the right of the farm households to use the land, but the ownership of the land is not attached to the farm households. For the case of SPK401 as an example, farm households are not allowed to sell their land to other people. The land can only be transferred to the heir of farm households and each farmer cannot hold more than eight hectares of land.

Table 2 presents the mean values of the selected variables for farms with and without full land ownership in each farm subgroup. We also test the mean difference between farms with and without full land ownership. Table 3 presents the same information while using a broader definition of land ownership, which is weak land ownership. We observe that the rice yields without full or weak land ownership are greater than those with full or weak land ownership at 5 percent level of significance across all types of farm. The amount of informal debt of farms with full land ownership is lower than those without full land ownership in small and mid-size farm subgroups at a 1 percent level of significance. We reveal that mid-size and large farm subgroups with weak land ownership have informal debt that is lower than those without weak land ownership at 1 and 5 percent level of significance, respectively, while using the definition of weak land ownership.

**Table 2.** Mean values of selected variables for farm types with/without full land ownership.

| | Small Farm | | | | Midsize Farm | | | | |
|---|---|---|---|---|---|---|---|---|---|
| **Variable** | **with** | **without** | **difference** | | **with** | **without** | **difference** | | **v** |
| *Outcome* | | | | | | | | | |
| Rice yield (kg/hectare) | 3,120 | 3,169 | −49 | ** | 2,906 | 3,061 | −155 | *** | 3 |
| Informal debt (USD) | 12.884 | 58.926 | −46.042 | *** | 30.983 | 73.977 | −42.994 | *** | 9 |
| *Principal characteristics* | | | | | | | | | |
| Male | 0.598 | 0.705 | −0.107 | *** | 0.59 | 0.67 | −0.08 | *** | 0 |
| Age (year) | 54.561 | 51.665 | 2.896 | *** | 55.232 | 52.483 | 2.749 | *** | 56 |
| Primary education | 0.048 | 0.032 | 0.016 | *** | 0.044 | 0.032 | 0.012 | *** | 0 |
| Single | 0.064 | 0.043 | 0.021 | *** | 0.053 | 0.037 | 0.016 | *** | 0 |
| Farmer group member | 0.133 | 0.122 | 0.011 | * | 0.143 | 0.124 | 0.019 | *** | 0 |
| Cooperative member | 0.105 | 0.101 | 0.004 | | 0.112 | 0.112 | 0 | | 0 |
| Village fund member | 0.018 | 0.031 | −0.013 | *** | 0.022 | 0.03 | −0.008 | *** | 0 |
| Agri. assoc. member | 0.003 | 0.005 | −0.002 | ** | 0.006 | 0.007 | −0.001 | | 0 |
| *Farm characteristics* | | | | | | | | | |
| Pct agri. Labor (%) | 0.713 | 0.721 | −0.008 | | 0.733 | 0.725 | 0.008 | | 0 |
| Work in agri. only | 0.632 | 0.65 | −0.018 | ** | 0.685 | 0.67 | 0.015 | * | 0 |
| Hire permanent labor | 0.469 | 0.509 | −0.04 | *** | 0.563 | 0.592 | −0.029 | *** | 0 |
| Hire temporary labor | 0.017 | 0.017 | 0 | | 0.021 | 0.02 | 0.001 | | 0 |
| Off−farm income | 0.336 | 0.248 | 0.088 | *** | 0.209 | 0.18 | 0.029 | *** | 0 |
| Ratio rice area | 0.569 | 0.453 | 0.116 | *** | 1.388 | 1.153 | 0.235 | *** | 2 |
| Ratio rice area^2 | 0.417 | 0.294 | 0.123 | *** | 2.32 | 1.73 | 0.59 | *** | 11 |
| Area harvested rice (hectare) | 0.968 | 0.969 | −0.001 | | 0.972 | 0.967 | 0.005 | * | 0 |
| Integrated agriculture | 7.326 | 9.994 | −2.668 | *** | 7.076 | 9.128 | −2.052 | *** | 7 |
| *Location characteristics* | | | | | | | | | |
| Rainfall (mm) | 1,503 | 1,421 | 82 | *** | 1,456 | 1,422 | 34 | *** | 1 |
| Temperature (°C) | 27.818 | 27.824 | −0.006 | * | 27.862 | 27.882 | −0.020 | *** | 27 |
| Municipal area | 0.302 | 0.250 | 0.051 | *** | 0.258 | 0.227 | 0.031 | *** | 0 |
| Irrigate (hectare) | 745,085 | 831,742 | −86,657 | *** | 724,605 | 831,528 | −106,922 | *** | 79 |
| No. observation | 7,646 | 5,612 | 2,034 | | 7,470 | 5,827 | 1,643 | | 6 |

Notes: Single, double, and triple asterisks (*, **, and ***) indicate significance at the 10%, 5%, and 1% level, respectively.

**Table 3.** Mean values of selected variables for farm types with/without weak land ownership.

| Variable | Small Farm | | | | Midsize Farm | | | | Large Farm | | | |
|---|---|---|---|---|---|---|---|---|---|---|---|---|
| | with | without | difference | | with | without | difference | | with | without | difference | |
| *Outcome* | | | | | | | | | | | | |
| Rice yield (kg/hectare) | 3,097 | 3,261 | −164 | *** | 2,869 | 3,256 | −387 | *** | 3,011 | 3,763 | −752 | *** |
| Informal debt (USD) | 25.884 | 50.251 | −24.367 | | 35.492 | 88.560 | −53.068 | *** | 87.577 | 179.936 | −92.359 | ** |
| *Principal characteristics* | | | | | | | | | | | | |
| Male | 0.617 | 0.717 | −0.100 | *** | 0.604 | 0.68 | −0.076 | *** | 0.633 | 0.682 | −0.049 | *** |
| Age (year) | 54.186 | 50.992 | 3.194 | *** | 54.76 | 52.047 | 2.713 | *** | 55.821 | 52.276 | 3.545 | *** |
| Primary education | 0.045 | 0.029 | 0.016 | *** | 0.041 | 0.033 | 0.008 | ** | 0.052 | 0.049 | 0.003 | |
| Single | 0.059 | 0.043 | 0.016 | *** | 0.048 | 0.039 | 0.009 | ** | 0.049 | 0.044 | 0.005 | |
| Farmer group member | 0.131 | 0.124 | 0.007 | | 0.137 | 0.127 | 0.01 | | 0.149 | 0.158 | −0.009 | |
| Cooperative member | 0.102 | 0.105 | −0.003 | | 0.113 | 0.111 | 0.002 | | 0.120 | 0.120 | 0 | |
| Village fund member | 0.021 | 0.031 | −0.010 | *** | 0.024 | 0.029 | −0.005 | * | 0.026 | 0.026 | 0 | |
| Agri. assoc. member | 0.004 | 0.005 | −0.001 | | 0.006 | 0.007 | −0.001 | | 0.005 | 0.007 | −0.002 | |
| *Farm characteristics* | | | | | | | | | | | | |
| Pct agri. Labor (%) | 0.721 | 0.703 | 0.018 | *** | 0.740 | 0.701 | 0.039 | *** | 0.749 | 0.692 | 0.057 | *** |
| Work in agri. only | 0.638 | 0.643 | −0.005 | | 0.686 | 0.658 | 0.028 | *** | 0.720 | 0.720 | 0 | |
| Hire permanent labor | 0.481 | 0.501 | −0.02 | ** | 0.571 | 0.588 | −0.017 | * | 0.618 | 0.658 | −0.040 | *** |
| Hire temporary labor | 0.018 | 0.015 | 0.003 | | 0.021 | 0.019 | 0.002 | | 0.022 | 0.02 | 0.002 | |
| Off−farm income | 0.314 | 0.256 | 0.058 | *** | 0.197 | 0.195 | 0.002 | | 0.130 | 0.094 | 0.036 | *** |
| Ratio rice area | 0.554 | 0.424 | 0.13 | *** | 1.358 | 1.091 | 0.267 | *** | 2.765 | 2.621 | 0.144 | *** |
| Ratio rice area^2 | 0.401 | 0.263 | 0.138 | *** | 2.241 | 1.578 | 0.663 | *** | 10.966 | 13.136 | −2.17 | |
| Area harvested rice (hectare) | 0.971 | 0.963 | 0.008 | ** | 0.974 | 0.96 | 0.014 | *** | 0.968 | 0.912 | 0.056 | *** |
| Integrated agriculture | 7.871 | 10.069 | −2.198 | *** | 7.58 | 9.045 | −1.465 | *** | 7.601 | 7.586 | 0.015 | |
| *Location characteristics* | | | | | | | | | | | | |
| Rainfall (mm) | 1,490 | 1,408 | 81 | *** | 1,451 | 1,414 | 37 | *** | 1,434 | 1,407 | 27 | *** |
| Temperature (°C) | 27.821 | 27.819 | 0.002 | | 27.861 | 27.900 | −0.039 | *** | 27.890 | 28.005 | −0.115 | *** |
| Municipal area | 0.294 | 0.242 | 0.052 | *** | 0.251 | 0.226 | 0.03 | *** | 0.213 | 0.205 | 0.008 | |
| Irrigate (hectare) | 747,294 | 876,729 | −129,435 | *** | 722,231 | 904,521 | −182,290 | *** | 782,612 | 1,092,597 | −309,985 | *** |
| No. observation | 9,727 | 3,531 | 6,196 | | 9,706 | 3,591 | 6,115 | | 7,835 | 4,590 | 3,245 | |

Notes: Single, double, and triple asterisks (*, **, and ***) indicate significance at the 10%, 5%, and 1% level, respectively.

The test of mean difference is also performed for factors that determine the potential outcomes. We observe that the mean values of several explanatory variables are different with statistical significance. For example, the proportions of male farmer and hire permanent labor with full and weak land ownership are lower than those without full and weak land ownership. Conversely, the mean age of the household head of farms with full and weak land ownership is greater than that without full and weak land ownership. The above findings show that we cannot estimate the impacts of land ownership on economic performance and the viability of rice farming by simply comparing the mean difference between farms with and without the land ownership without addressing potential selection bias.

## 3. Results

This section begins with the propensity score estimation, and then reveals the estimated effects of land ownership on the economic performance and viability of rice farming. Matching quality and robustness checks are performed and then discussed in the last section for the accuracy of estimation.

### 3.1. Propensity Score Estimation

In the first stage of the PSM technique, we estimated logit models for each type of farm by regressing each binary treatment variable on the multi-dimensional vector of covariates. Tables 4 and 5 provide the parameter estimates that were obtained from models corresponding to the treatment variables of "full land ownership" and "weak land ownership", respectively. All six models performed well according to the percentage of correct predictions, although the models have low values of the Pseudo $R^2$, which generally exist with the application while using the cross-sectional data, as found in several search papers [8,23].

We found similarities among the factors that determine the probability of having full or weak land ownership across three types of farm. We observed that farm households are more likely to have full or weak land ownership if their household heads are female; have a higher age; graduated at least from primary education; have a single marital status; conduct non-integrated agriculture; and, receive a higher temperature. The probability of having land ownership is positively correlated to the ratio of rice planting area to the total cropland for small and medium farm sizes, with an inverted U-shape relationship for the large farm size.

**Table 4.** Estimated coefficients from logit models classified by farm type: full land ownership.

| Variables | Small Coef. | | SE | Midsize Coef. | | SE | Large Coef. | | SE |
|---|---|---|---|---|---|---|---|---|---|
| *Principal characteristics* | | | | | | | | | |
| Male | −0.303 | *** | 0.041 | −0.228 | *** | 0.038 | −0.197 | *** | 0.040 |
| Age | 0.024 | *** | 0.002 | 0.021 | *** | 0.002 | 0.023 | *** | 0.002 |
| Primary education | 0.474 | *** | 0.101 | 0.462 | *** | 0.103 | 0.550 | *** | 0.089 |
| Single | 0.462 | *** | 0.086 | 0.418 | *** | 0.093 | 0.537 | *** | 0.092 |
| Farmer group member | 0.226 | *** | 0.058 | 0.247 | *** | 0.055 | 0.125 | ** | 0.053 |
| Cooperative member | 0.236 | *** | 0.062 | 0.104 | * | 0.059 | 0.047 | | 0.058 |
| Village fund member | −0.351 | *** | 0.124 | −0.174 | | 0.115 | −0.050 | | 0.120 |
| Agri. assoc. member | −0.522 | * | 0.304 | −0.055 | | 0.243 | −0.284 | | 0.257 |
| *Farm characteristics* | | | | | | | | | |
| Pct agri. labor | −0.143 | ** | 0.065 | −0.037 | | 0.065 | 0.233 | *** | 0.066 |
| Work in agri. only | −0.016 | | 0.043 | 0.015 | | 0.042 | 0.032 | | 0.044 |
| Hire permanent labor | −0.084 | ** | 0.039 | −0.008 | | 0.038 | −0.119 | *** | 0.040 |
| Hire temporary labor | 0.075 | | 0.159 | −0.011 | | 0.133 | 0.191 | | 0.138 |
| Off−farm income | 0.282 | *** | 0.047 | 0.058 | | 0.051 | 0.110 | * | 0.062 |
| Ratio rice area | 0.522 | *** | 0.058 | 0.123 | *** | 0.011 | 0.245 | *** | 0.029 |
| Ratio rice area^2 | | | | | | | −0.015 | *** | 0.003 |
| Area harvested rice | −0.046 | | 0.116 | 0.061 | | 0.117 | 0.534 | *** | 0.106 |
| Integrated agriculture | −0.078 | *** | 0.004 | −0.068 | *** | 0.005 | −0.044 | *** | 0.004 |
| *Location characteristics* | | | | | | | | | |
| Rainfall | 0.001 | *** | $1.00 \times 10^{-4}$ | 0.001 | *** | $1.00 \times 10^{-4}$ | $6.00 \times 10^{-4}$ | *** | $2.00 \times 10^{-4}$ |
| Temperature | 1.225 | *** | 0.149 | 1.283 | *** | 0.183 | 0.955 | *** | 0.221 |
| Municipal area | 0.233 | *** | 0.043 | 0.150 | *** | 0.044 | 0.070 | | 0.047 |
| Irrigate | $-9.46 \times 10^{-7}$ | *** | $8.68 \times 10^{-8}$ | $-1.43 \times 10^{-6}$ | *** | $1.00 \times 10^{-7}$ | $-1.47 \times 10^{-6}$ | *** | $1.26 \times 10^{-7}$ |
| Constant | −35.435 | | 4.125 | −36.693 | | 4.985 | −28.309 | | 5.913 |
| Pseudo $R^2$ | 0.103 | | | 0.080 | | | 0.078 | | |
| % Correctly predicted | 67.48% | | | 65.37% | | | 64.72% | | |
| No. observations | 13,258 | | | 13,297 | | | 12,425 | | |

Notes: Single, double, and triple asterisks (*, **, and ***) indicate significance at the 10%, 5%, and 1% level, respectively.

**Table 5.** Estimated coefficients from logit models classified by farm type: weak land ownership.

| Variables | Small Coef. | | SE | Midsize Coef. | | SE | Large Coef. | | SE |
|---|---|---|---|---|---|---|---|---|---|
| *Principal characteristics* | | | | | | | | | |
| Male | −0.258 | *** | 0.046 | −0.181 | *** | 0.045 | −0.206 | *** | 0.044 |
| Age | 0.028 | *** | 0.002 | 0.020 | *** | 0.002 | 0.026 | *** | 0.002 |
| Primary education | 0.537 | *** | 0.120 | 0.456 | *** | 0.119 | 0.361 | *** | 0.096 |
| Single | 0.408 | *** | 0.098 | 0.325 | *** | 0.107 | 0.507 | *** | 0.100 |
| Farmer group member | 0.198 | *** | 0.064 | 0.194 | *** | 0.063 | 0.050 | | 0.057 |
| Cooperative member | 0.162 | ** | 0.070 | 0.124 | * | 0.067 | −0.009 | | 0.061 |
| Village fund member | −0.182 | | 0.124 | −0.134 | | 0.130 | 0.007 | | 0.126 |
| Agri. assoc. member | −0.335 | | 0.329 | −0.056 | | 0.272 | −0.183 | | 0.271 |
| *Farm characteristics* | | | | | | | | | |
| Pct agri. labor | 0.048 | | 0.071 | 0.221 | *** | 0.073 | 0.391 | *** | 0.070 |
| Work in agri. only | −0.026 | | 0.048 | 0.031 | | 0.048 | 0.050 | | 0.048 |
| Hire permanent labor | −0.036 | | 0.043 | 0.041 | | 0.043 | −0.030 | | 0.043 |
| Hire temporary labor | 0.198 | | 0.181 | 0.048 | | 0.154 | 0.024 | | 0.150 |
| Off−farm income | 0.154 | *** | 0.052 | −0.105 | * | 0.057 | 0.083 | | 0.069 |
| Ratio rice area | 0.636 | *** | 0.070 | 0.189 | *** | 0.014 | 0.347 | *** | 0.037 |
| Ratio rice area^2 | | | | | | | −0.023 | *** | 0.004 |
| Area harvested rice | 0.166 | | 0.120 | 0.128 | | 0.123 | 0.623 | *** | 0.115 |
| Integrated agriculture | −0.043 | *** | 0.004 | −0.032 | *** | 0.004 | −0.012 | *** | 0.004 |
| *Location characteristics* | | | | | | | | | |
| Rainfall | 0.001 | *** | $1.00 \times 10^{-4}$ | $5.00 \times 10^{-4}$ | *** | $1.00 \times 10^{-4}$ | $2.00 \times 10^{-4}$ | | $2.00 \times 10^{-4}$ |
| Temperature | 2.511 | *** | 0.166 | 2.644 | *** | 0.191 | 1.826 | *** | 0.314 |
| Municipal area | 0.251 | *** | 0.049 | 0.151 | *** | 0.050 | 0.006 | | 0.051 |
| Irrigate | $-1.94 \times 10^{-6}$ | *** | $1.00 \times 10^{-7}$ | $-2.54 \times 10^{-6}$ | *** | $1.08 \times 10^{-7}$ | $-2.39 \times 10^{-6}$ | *** | $1.82 \times 10^{-7}$ |
| Constant | −70.177 | | 4.563 | −72.901 | | 5.206 | −51.400 | | 8.345 |
| Pseudo R$^2$ | 0.109 | | | 0.106 | | | 0.126 | | |
| % Correctly predicted | 75.86% | | | 75.29% | | | 70.61% | | |
| No. observations | 13,258 | | | 13,297 | | | 12,425 | | |

Notes: Single, double, and triple asterisks (*, **, and ***) indicate significance at the 10%, 5%, and 1% level, respectively.

Mixed findings are revealed between these two treatment variables. We found that an increase in the percent of agricultural labor to total labor ratio in the household diminishes the likelihood of having full land ownership of the small farm households, while it tends to enhance the likelihood of having full or weak land ownership for the large farm households. Moreover, farm households that hire permanent labor are likely to have a lower probability of having full land ownership for small and large farms. Participating as a member of a farmer group or cooperative enhances the probability of having full or weak land ownership in small and medium farm households. Finally, small farm households that have the largest source of income from off-farm income and are located in the municipal area are likely to have both full and weak land ownership.

### 3.2. Effects of Land Ownership on the Economic Performance and Viability of Rice Farming

A propensity score from the logit model that was described in the previous section was derived to identify the predicted probability of having full or weak land ownership for each individual farm. Farms with the ownership status were then matched to farms without the ownership status that was based on the propensity scores while using the nearest-neighbour matching with one to five (NN5) and one to ten (NN10), Gaussian kernel matching, and radius matching with the caliper of 0.01 and 0.02. The use of multiple matching estimators is a useful robustness check and it provided a means of observing the sensitivity of estimated ATTs.

This study employed the trimming approach to estimate ATTs for each potential outcome across all three types of farm, as shown in Table 6, because of the several advantages that the trimming approach provides as compared to the common support approach [29], as discussed in the section on the materials and methods. Standard errors are reported in parentheses under each estimated treatment effect while using the bootstrap method, with 50 replications, except for the nearest-neighbour matching, in which we calculated the analytical standard errors that were suggested by [33].

**Table 6.** Average treatment effects (ATTs) of outcomes while using the trimming approach.

| Sample | NN 5 | | NN 10 | | Kernel | | Radius 0.01 | | Radius 0.02 | |
|---|---|---|---|---|---|---|---|---|---|---|
| | | | | | **Matching Algorithms** | | | | | |
| | | | | | *Outcome: Rice yield (kilograms/hectare)* | | | | | |
| *Treatment: Full land ownership* | | | | | | | | | | |
| Small | 115.789 | *** | 117.504 | *** | 126.129 | *** | 125.321 | *** | 127.414 | *** |
| | (26.493) | | (25.461) | | (25.715) | | (26.0156) | | (25.934) | |
| Midsize | 70.707 | *** | 55.129 | ** | 51.926 | ** | 54.125 | ** | 54.293 | ** |
| | (23.469) | | (22.852) | | (24.856) | | (25.121) | | (25.039) | |
| Large | 47.863 | * | 46.519 | * | 34.809 | * | 40.649 | * | 39.586 | ** |
| | (24.734) | | (24.059) | | (28.293) | | (28.606) | | (28.539) | |
| *Treatment: Weak land ownership* | | | | | | | | | | |
| Small | 66.785 | ** | 72.574 | ** | 65.590 | ** | 70.406 | ** | 68.504 | ** |
| | (30.547) | | (29.894) | | (31.066) | | (31.477) | | (31.383) | |
| Midsize | −12.403 | | −16.371 | | −28.512 | | −19.321 | | −22.981 | |
| | (27.285) | | (26.887) | | (31.766) | | (32.340) | | (32.199) | |
| Large | −30.793 | | −23.196 | | −20.012 | | −23.199 | | −17.887 | |
| | (28.644) | | (27.031) | | (34.180) | | (35.797) | | (35.379) | |
| | | | | | *Outcome: Informal debt (USD)* | | | | | |
| *Treatment: Full land ownership* | | | | | | | | | | |
| Small | −17.199 | ** | −16.972 | ** | −24.324 | ** | −24.877 | ** | −24.200 | *** |
| | (7.596) | | (7.345) | | (24.464) | | (24.862) | | (24.749) | |
| Midsize | −31.393 | ** | −36.194 | ** | −37.819 | ** | −37.720 | ** | −37.276 | *** |
| | (12.504) | | (14.316) | | (16.421) | | (16.606) | | (16.549) | |
| Large | −2.590 | | 2.058 | | −21.318 | | −18.705 | | −19.672 | |
| | (35.026) | | (33.304) | | (43.083) | | (43.491) | | (43.394) | |
| *Treatment: Weak land ownership* | | | | | | | | | | |
| Small | −21.936 | | −14.939 | | −15.280 | | −14.070 | | −14.133 | |
| | (20.005) | | (16.658) | | (18.524) | | (18.701) | | (18.663) | |
| Midsize | −44.681 | ** | −37.983 | ** | −38.735 | *** | −37.440 | *** | −36.909 | *** |
| | (19.026) | | (15.721) | | (23.438) | | (23.922) | | (23.801) | |
| Large | −82.945 | | −80.713 | | −78.404 | | −75.390 | | −74.309 | |
| | (64.694) | | (70.039) | | (54.785) | | (57.355) | | (56.681) | |

Notes: Single, double, and triple asterisks (*, **, and ***) indicate significance at the 10%, 5%, and 1% level, respectively. Standard errors are reported in parentheses. The standard errors for all matching algorithms were estimated using bootstrapping with 50 replications, except for the oversampling (NN5 and NN10), for which we used the analytical standard error suggested by [33]

While considering the potential outcome of the rice yield per hectare, we revealed that the estimated ATTs for the treatment variable of full land ownership are positive and statistically significant across all of the matching techniques and all types of farm. The small-size farms obtain the largest benefit of having full land ownership, with an increase in rice yield between 115.789 and 127.414 kg/hectare. Full land ownership enhances the rice yield of midsize and large farm subgroups, which ranged from 51.926 to 70.707 kg/hectare and 34.809 to 47.863 kg/hectare, respectively. While considering the effect of weak land ownership on the rice yield, we found that weak land ownership only enhances the rice yield of the small-size farm subgroup, with an increased yield of 65.590–72.574 kg/hectare. We also observed that the effect of full land ownership on the rice yield is greater than that of weak land ownership.

We observed that the estimated ATTs for full land ownership are negative and statistically significant at a 5% level across all matching techniques of small and midsize farm subgroups while considering the effect of land ownership on the informal debt of farm households. These findings imply that informal debt can be reduced by encouraging farm households to have full land ownership. The informal debt of small and midsize farm subgroups with full land ownership was estimated to reduce between $16.972 and $24.877 per farm and $31.393 and $37.819 per farm, respectively. On the other hand, having weak land ownership reduced the informal debt of midsize farms, ranging from $36.909 to $44.681 per farm.

As a robustness check, we estimated the average treatment effects (ATT) of the rice yield and informal debt while using the common support approach, as shown in Table S1 in the supplemental online material, and found that the estimated ATT were close to the results that were obtained with the trimming approach, as demonstrated in Table 6.

### 3.3. Matching Quality and Robustness Checks

We assessed the quality of all matching algorithms using the mean standardized bias and Pseudo $R^2$ for more robustness checks, as shown in Table S2 for the treatment variable of full land ownership and Table S3 for the treatment variable of weak land ownership. We found that the mean standardized bias and Pseudo $R^2$ of all matching algorithms dropped sharply after matching, as shown in Table S2, thus implying that full land ownership enhances the rice yield of all types of farm and helps to reduce the informal debt of small and midsize farms. Kernel matching provides the highest quality of matching for small and large farm subgroups, while the radius matching with the caliper of 0.01 provides the highest quality of matching for the midsize farm subgroup.

The mean standardized bias and Pseudo $R^2$ of all matching algorithms also dropped sharply after matching, as presented in Table S3, which implies that weak land ownership enhances the rice yield of a small size farm and helps to reduce the informal debt of midsize farms. The nearest-neighbor matching with one to five (NN5) provides the highest quality of matching for the small farm subgroup, while the radius matching with the caliper of 0.02 provides the highest quality of matching for the midsize farm subgroup.

This study also performed the covariate balancing test before and after matching as another robustness check, as suggested by the literature. Overall, we found that, before matching, the mean values of almost all explanatory variables between farms with and without full land ownership were different, with statistical significance (Table S4). However, after matching, the mean values of only 2–3 explanatory variables were different with statistical significance, implying that the increase in rice yield and decrease in informal debt were likely generated from the full land ownership, similar to the results of the previous robustness check while using the mean standardized bias and Pseudo $R^2$. Similar findings were revealed for the treatment variable of weak land ownership that is shown in Table S5, although the quality of matching was generally lower than the treatment variable of full land ownership presented in Table S4.

Finally, PSM relied on the conditional independence assumption (CIA), as discussed in Section 2. This means that the estimated ATTs that were based on matching were unbiased if all relevant covariates were included in the model, which is a rather restrictive assumption. A common concern of matching models is that they may fail to account for a relevant covariate(s) that is not observable

to researchers. Consequently, we found that the rice yield of the large farm subgroup for the treatment variable of full land ownership might be sensitive to the violation of CIA by using Rosenbaum bounds with one-to-one matched pairs, as proposed by [34] (Table 7).

Therefore, we can conclude that full land ownership enhances the rice yield of both small and midsize farms, while weak land ownership only improves the rice yield of small farms. With the full land ownership, the increased rice yield of the small size farm subgroup is greater than that of the midsize farm subgroup. Moreover, full land ownership reduces the informal debt of small and midsize farms, while weak land ownership only reduces the informal debt of the midsize farms. The midsize farm subgroup seems to obtain the largest benefit from the reduction in informal debt as compared to the small size farms. Section 4 provides a discussion of these findings in detail.

**Table 7.** Sensitivity analysis with Rosenbaum bounds for treatment variables.

| | Gamma | 1 | 1.05 | 1.1 | 1.15 | 1.2 | 1.25 | 1.3 | 1.35 | 1.4 | 1.45 | 1.5 |
|---|---|---|---|---|---|---|---|---|---|---|---|---|
| | | | | | *Treatment: Full land ownership* | | | | | | | |
| *Outcome: Rice yield (kilograms/hectare)* | | | | | | | | | | | | |
| Small | sig+ | $9.70 \times 10^{-9}$ | $7.30 \times 10^{-5}$ | 0.019 | 0.339 | 0.879 | 0.996 | 0.999 | 1 | 1 | 1 | 1 |
| | sig− | $9.70 \times 10^{-9}$ | $5.10 \times 10^{-14}$ | 0 | 0 | 0 | 0 | 0 | 0 | 0 | 0 | 0 |
| Midsize | sig+ | $5.30 \times 10^{-5}$ | 0.019 | 0.366 | 0.904 | 0.998 | 0.999 | 1 | 1 | 1 | 1 | 1 |
| | sig− | $5.30 \times 10^{-5}$ | $6.30 \times 10^{-9}$ | $5.70 \times 10^{-14}$ | 0 | 0 | 0 | 0 | 0 | 0 | 0 | 0 |
| Large | sig+ | 0.043 | 0.464 | 0.929 | 0.998 | 0.999 | 1 | 1 | 1 | 1 | 1 | 1 |
| | sig− | 0.043 | $3.96 \times 10^{-4}$ | $4.40 \times 10^{-7}$ | $7.40 \times 10^{-11}$ | $2.20 \times 10^{-15}$ | 0 | 0 | 0 | 0 | 0 | 0 |
| *Outcome: Informal debt (USD)* | | | | | | | | | | | | |
| Small | sig+ | $1.01 \times 10^{-4}$ | $2.40 \times 10^{-5}$ | $5.40 \times 10^{-6}$ | $1.10 \times 10^{-6}$ | $2.40 \times 10^{-7}$ | $4.70 \times 10^{-8}$ | $8.90 \times 10^{-9}$ | $1.70 \times 10^{-9}$ | $3.00 \times 10^{-10}$ | $5.40 \times 10^{-11}$ | $9.40 \times 10^{-12}$ |
| | sig− | $1.01 \times 10^{-4}$ | $3.79 \times 10^{-4}$ | 0.001 | 0.003 | 0.008 | 0.016 | 0.031 | 0.055 | 0.089 | 0.135 | 0.192 |
| Midsize | sig+ | $6.80 \times 10^{-7}$ | $8.10 \times 10^{-8}$ | $9.00 \times 10^{-9}$ | $9.50 \times 10^{-10}$ | $9.50 \times 10^{-11}$ | $9.10 \times 10^{-12}$ | $8.40 \times 10^{-13}$ | $7.40 \times 10^{-14}$ | $6.30 \times 10^{-15}$ | $5.60 \times 10^{-16}$ | 0 |
| | sig− | $6.80 \times 10^{-7}$ | $4.80 \times 10^{-6}$ | $2.60 \times 10^{-5}$ | $1.16 \times 10^{-4}$ | $4.26 \times 10^{-4}$ | 0.001 | 0.004 | 0.009 | 0.018 | 0.035 | 0.062 |
| | | | | | *Treatment: Weak land ownership* | | | | | | | |
| *Outcome: Rice yield (kilograms/hectare)* | | | | | | | | | | | | |
| Small | sig+ | $3.60 \times 10^{-9}$ | $6.80 \times 10^{-5}$ | 0.026 | 0.442 | 0.942 | 0.999 | 1 | 1 | 1 | 1 | 1 |
| | sig− | $3.60 \times 10^{-9}$ | $4.30 \times 10^{-15}$ | 0 | 0 | 0 | 0 | 0 | 0 | 0 | 0 | 0 |
| *Outcome: Informal debt (USD)* | | | | | | | | | | | | |
| Midsize | sig+ | $7.50 \times 10^{-9}$ | $4.10 \times 10^{-10}$ | $2.00 \times 10^{-11}$ | $9.30 \times 10^{-13}$ | $3.90 \times 10^{-14}$ | $1.60 \times 10^{-15}$ | $1.10 \times 10^{-16}$ | 0 | 0 | 0 | 0 |
| | sig− | $7.50 \times 10^{-9}$ | $1.10 \times 10^{-7}$ | $1.10 \times 10^{-6}$ | $8.20 \times 10^{-6}$ | $4.80 \times 10^{-5}$ | $2.24 \times 10^{-4}$ | $8.50 \times 10^{-4}$ | 0.003 | 0.007 | 0.018 | 0.037 |

[a] Gamma, log odds of differential assignment due to unobserved factors; sig+, upper bound significance level; sig-, lower bound significance level. The boxed numbers indicate the critical level of the strength of the effect of Gamma for each of the dependent variables.

## 4. Discussion

There are two general explanations for the findings that were demonstrated in the previous section. The first explanation derives from the liquidity constraint that farms face when they want to spend money on farm investment, while the second explanation comes from the motivation to do farming of farm households.

Land with full ownership will generally have a higher value than land with weak ownership from the perspective of the liquidity constraint. Additionally, farms with full land ownership can have a higher bank loan for farm investment than those with weak land ownership. Moreover, farms without land ownership cannot access bank loans for farm investment. Empirically, the land with full ownership in Thailand (i.e., title deed and NS3) receives bank loans of at least 80 percent of the land value [35], while the land with weak ownership (i.e., SPK401 NK NS2, and SK1) obtains bank loans of, at most, 50 percent of the land value [36]. Consequently, farms with weak or no land ownership will face more liquidity constraints than those with full land ownership. The liquidity constraint might be the most important factor that enhances the economic performance and viability of small and midsize farms when compared to large-size farms.

The land ownership can also affect the motivation to do farming of farm households. The land with weak land ownership that is defined in this study cannot be sold to other people, but it can be transferred to the heir of the farm household, according to the law. As a result, if the farm households do not want to do farming, they will not invest in farm modernization, and land with weak land ownership will finally be left desolate. Similar to the discussion on the potential outcome of the rice yield, the liquidity constraint and the motivation to do farming are also two key factors determining the reduction of informal debt of farm households.

Our findings are similar to the results from [12], who interviewed 950 farmers in four provinces of Pakistan and revealed that farmers growing wheat, rice, and cotton with secure land rights had higher crop yields and lower poverty levels as compared with farmers without secure land rights. [11,15] also revealed that land ownership was negatively associated with inefficiency. In Thailand, our results at the national level also confirmed the findings of [10,16], who used the cross-sectional farm-level data that were collected in 1984/85 from interviewing 200 farmers in three provinces of Thailand (i.e., Lopburi, Nakhon Ratchasima, and Khon Kaen) and revealed that land ownership enhances farm productivity. The current study provides several new findings that there are heterogeneous effects of full and weak land ownership and different farm types also receive different impacts of land ownership on the economic performance and viability of rice farming, as revealed in the previous section.

## 5. Conclusions

This article evaluates the impacts of land ownership on the economic performance and viability of rice farming and explores whether they are heterogeneous across different farm types (i.e., small, midsize, and large farms). The PSM technique was utilized to address the possible selection bias with a constructed farm-level dataset. This study categorizes land ownership into two types: 1) "full land ownership", capturing all characteristics of the well-defined property right structure in economic theory consisting of exclusivity, transferability, and enforceability, and "weak land ownership", only capturing characteristics of exclusivity and enforceability, to thoroughly understand the effect of land ownership.

We have revealed that full land ownership enhances the rice yield of small and midsize farm subgroups, which range from 115.789 to 127.414 kg/hectare and from 51.926 to 70.707 kg/hectare, respectively. On the other hand, weak land ownership only enhances the rice yield of the small farm subgroup, with an increased yield of 65.590–72.574 kg/hectare. Full land ownership also helps to reduce the informal debt of small and midsize farm subgroups by between $16.972 and $24.877 per farm and $31.393 and $37.819 per farm, respectively. On the other hand, having weak land ownership helps to reduce the informal debt of the midsize farms only, ranging from $36.909 to $44.681 per farm.

Policy makers should encourage small and midsize farm households to have at least weak land ownership to improve the rice yield and reduce the informal debt of farm households. Encouraging

small and midsize farm households to have full land ownership instead of weak land ownership will provide the greatest benefits to farm households and an efficient use of land. For example, policy makers may relax the restriction of SPK401 from "not allowing to sell" to "allowing to sell". This relaxation will promote land with an SPK401 certificate to have a well-defined property right structure.

Policy makers can learn from the experience of the farmland preservation program's so-called "purchase of development rights (PDR)" that is widely used in the United States. This program affords the permanent protection of farmland from conversion to non-agricultural development. Participation in a PDR program requires a landowner to forfeit the right to develop farmland for non-agricultural purposes and a conservation easement is placed on the land. In exchange, the landowner receives a monetary payment (or, in some cases, a tax incentive) and retains ownership and all other land rights.

This program is an attractive public policy from a property rights perspective, because landowner equity is protected due to the voluntary and compensatory nature of program participation, thus avoiding political and legal challenges to the constitutionality of regulatory-based land management approaches [37–38]. In addition to the permanence of farmland protections, the program offers several other advantages. It is theorized that the removal of farm investments will be spurred by the infusion of easement monies and might help to reverse the "impermanence syndrome", which [39] identified as afflicting urban-influenced farms. Moreover, restricting future non-agricultural development options should, again in theory, reduce the cost of farmland.

**Supplementary Materials:** The following are available online at www.mdpi.com/2073-445X/9/3/71/s1: Table S1: Average treatment effects on outcomes using the common support approach; Table S2: Matching quality indicators with the trimming approach corresponding to the potential outcomes with the treatment variable of full land ownership; Table S3: Matching quality indicators with the trimming approach corresponding to the potential outcomes with the treatment variable of weak land ownership; Table S4: Balancing test for the mean difference before and after matching corresponding to the potential outcomes with the treatment variable of full land ownership; Table S5: Balancing test for the mean difference before and after matching corresponding to the potential outcomes with the treatment variable of weak land ownership.

**Author Contributions:** All authors have read and agree to the published version of the manuscript. Conceptualization, A.P., W.A. and A.K.; Methodology, A.P., W.A. and A.K.; Software, A.P. and W.A.; Validation, W.A. and A.K.; Formal Analysis, A.P. and W.A.; Data Curation, A.P.; Writing – Original Draft Preparation, A.P.; Writing – Review & Editing, W.A.; Visualization, A.P.; Supervision, W.A. All authors have read and agreed to the published version of the manuscript.

**Funding:** This research received no external funding.

**Conflicts of Interest:** The authors declare no conflict of interest.

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
