# Peer review of "Impacts of Land Ownership on the Economic Performance and Viability of Rice Farming in Thailand"

_land, doi:10.3390/land9030071_

Round 1

Reviewer 1 Report

First, it has been a bit hard to properly comment the manuscript without line numbering... Please, do use it next time to facilitate the reviewer's task. 

I will write to you my comments as I wrote down them during my pen-on-paper revision. 

I appreciated the abstract, it is very exhaustive and it shows great efficacy. In the 2-4 lines of the paper (Introduction) it seems to be a bit straight statement... But I can guess it is motivated by what follows (info, references, data...)

Page 2, 2nd paragraph... There is a typo... "ref. [14]"

4th paragraph, same page... You say something crucial about the data set but no info are provided on HOW you did build it... This should be clarified. 

Page 3... I do appreciate the state-of-the-art about the straight subject of the topic but I could appreciate an up-to-date review of the application of PSM, PSM+DID, etc in the field of agri-environmental economics... At least on the most up-to-date published papers that deal with the challenge of policy-evaluation, impact of "treatment" in agri-environmental field, etc.

Same page, last lines... I do worry a bit about the un-official typology classifying. What about the references that support your choice in dividing them?! Is it not too much "data-driven", your choice? 

Section 2.2, you talk again about data and data set construction... See the previous comment, please. 

Table 2. I would suggest to test for the differences in means between the groups/categories and show/comment the results.

Section 3.1, 2nd paragraph, lines 2 and 3: i do not get the sense of the statement... There are some typos, perhaps?!

Table 4... The pseudo-Rseem odd... Any comments on them?

Section 4, 2nd paragraph. I would prefer to have a citation/reference rather than "empirically"... Is it possible? 

List of references seems to be okay but rather dated and there are no recent works in the field of agricultural economics that use the method... Any chance to improve it? With a broad discussion of the most up-to-date works that deal with the challenge of impact evaluation on agricultural holdings? 

>>> I really appreciated the manuscript, it is well-written, efficient and efficace. I would rather change a bit the methodological section, moving out from the scholar structure proposed and scaling down the method on the research subject. 

There are few issues to be solved, but the manuscript is really well done and it is enjoyable to read. 

Thank you for the possibility you gave to me to read it <<<

Author Response

We addressed all of your comments. Please see them in the attached file.

Reviewer 2 Report

The authors have used propensity scores matching to evaluate impacts of land ownership on the economic performance and viability of rice farming in Thailand. They showed sufficient background on used methodology and they included relevant references.
The research design was appropriate and data set was exaustive of country'reality.
The propensity scores matching is adequately described and the results clearly presented.

The conclusions, indirizzate to policy makers, were good supported by the results founded.In all tables and in text, it is better to use only three decimal places for numbers.

Two only recomendation:

1) In Materials and methods, it is better to indicate the software (package and its version) that was used.

2) It is better to indicate the numbers with only trhee cifre decimali.

Reviewer 3 Report

The MS is well structured and well written and the subject is interesting and relevant.

However, there are some issues that would be important to clarify or improve, such as:

There are compared two types of land ownership but, in fact it seems to be 3 types: no ownership / weak ownership / full ownership. Why not to use all the 3 ownership levels?

I did not understand some statements. For example, the paragraph before the discussion says: "we can conclude that full land ownership enhances the rice yield of small and midsize farms only" and after "we can conclude that weak land ownership increases the rice yield of the small farm". So, both total and weak land ownership increase production on small farms? If so in relation to what?

For most numerical values, there are many decimal places, two are sufficient. Units are missing from some variables in the tables.

See also the comments in the pdf file.
